Effects of sound exposure on the growth and intracellular macromolecular synthesis of E. coli k-12

Gu Shaobin shaobingu@haust.edu.cn
Zhang Yongzhu
Wu Ying
College of Food and Bioengineering, Henan University of Science and Technology , Luoyang Henan , China
Visweswariah Sandhya
Electronic publication date: 2016 Apr 7
Publication date: 2016
Volume: 4
Electronic Location ID: e1920
Received 2015 Dec 5; Accepted 2016 Mar 19
Copyright: ©2016 Gu et al.
Copyright year: 2016
Copyright holder: Gu et al.
License: This is an open access article distributed under the terms of the Creative Commons Attribution License, which permits unrestricted use, distribution, reproduction and adaptation in any medium and for any purpose provided that it is properly attributed. For attribution, the original author(s), title, publication source (PeerJ) and either DOI or URL of the article must be cited.
License URL: https://creativecommons.org/licenses/by/4.0/

Keywords: Sound exposure, E. coli K-12, Biomass, Cellular morphology

Funding: National Natural Science Foundation of China U1304307 Young Core Instructor Foundation from the Education Commission of Henan Province, China 2014GGJS-056 This work was supported by National Natural Science Foundation of China (Grant No. U1304307) and the Young Core Instructor Foundation from the Education Commission of Henan Province, China (Grant No. 2014GGJS-056). The funders had no role in study design, data collection and analysis, decision to publish, or preparation of the manuscript.

==============================
Microbes, as one of the primary producers of the biosphere, play an important role in ecosystems. Exploring the mechanism of adaptation and resistance of microbial population to various environmental factors has come into focus in the fields of modern microbial ecology and molecular ecology. However, facing the increasingly serious problem of acoustic pollution, very few efforts have been put forth into studying the relation of single cell organisms and sound field exposure. Herein, we studied the biological effects of sound exposure on the growth of E. coli K-12 with different acoustic parameters. The effects of sound exposure on the intracellular macromolecular synthesis and cellular morphology of E. coli K-12 were also analyzed and discussed. Experimental results indicated that E. coli K-12 exposed to sound waves owned a higher biomass and a faster specific growth rate compared to the control group. Also, the average length of E. coli K-12 cells increased more than 27.26%. The maximum biomass and maximum specific growth rate of the stimulation group by 8000 Hz, 80dB sound wave was about 1.7 times and 2.5 times that of the control group, respectively. Moreover, it was observed that E. coli K-12 can respond rapidly to sound stress at both the transcriptional and posttranscriptional levels by promoting the synthesis of intracellular RNA and total protein. Some potential mechanisms may be involved in the responses of bacterial cells to sound stress.

Introduction

The sound wave is a key component of environmental factors (Levin, 1995). In the natural world almost all living organisms are “immersed” in a variety of sound waves and interact with them (Dãnet, 2005). Sound waves are roughly classified into three regimes by its frequency: infrasound (10−4–20 Hz), audible sound (20 × 104 Hz) and ultrasound (2 × 104–1012 Hz). Ultrasound-induced biological effects and its biophysical mechanisms have been extensively investigated in recent decades (Leighton, 2007; William & O’Brien, 2007). It has already been successfully combined with biotechnology with the aim of enhancing the efficiency of bioprocesses (Rokhina, Lens & Virkutyte, 2009). Moreover, ultrasound is also widely used in medicine as both diagnostic and therapeutic tools (Whittingham, 2007). Although infrasound-induced bioeffects and its potential mechanism are still not clearly understood (Leventhall, 2007), some significant progresses have been made in the application field, e.g., infrasound diagnosis and therapeutic infrasound (Anastassiades & Petounis, 1976; Yount et al., 2004).

However, with the rapid development of industrialization and urbanization, acoustic pollution has increasingly become the focus of attention. After air and aquatic pollution, noise pollution is considered third-largest public hazard in the modern cities. Long-term exposures to noise will cause diseases such as heart vascular disease, cognitive dysfunction, sleep disorders, and tinnitus (Fritschi et al., 2011). For each 10 dB of added sound exposure, the stroke rate will increase by 1.14% (Sørensen et al., 2011), and the incidence of myocardial infarction will enhance by 1.12% (Sørensen et al., 2012). Moreover, a recently published study by Chan & Blumstein, 2012 shows that anthropogenic noises may modify critical ecological services in a woodland ecosystem and, by enhancing pollination and disrupted seed dispersal, expands the impact of noise into a new discipline—community ecology. Microbes, as one of the primary producers of the biosphere, play an important role in ecosystems, whether long-term sound exposure will change their physiological and ecological behavior, and how these changes affect themselves and the whole ecological system. To explore the mechanism of adaptation and resistance of microbial population to sound exposure, we examined the effects of sound exposure on the growth of E. coli K-12 with different acoustic parameters. The effects of sound exposure on the intracellular macromolecular synthesis and cellular morphology of E. coli K-12 were also analyzed and discussed.

Materials and Methods

E. coli strain

E. coli K-12 (4401, from The Coli Genetic Stock Center) was first cultured in LB slant agar medium at 37 °C for 24 h. Then, cells were expanded in a 250 ml conical flask containing 100 ml of LB liquid medium with agitation of 180 rpm on a rotary shaking incubator at 37 °C for 10 h.

Sound wave load apparatus

Sound exposure test were performed in the experimental installations (Fig. 1). This equipment was composed of the sound wave generating unit and sound wave load chamber. The former contained a waveform generator and the amplifying circuit. The signals produced by the waveform generator were amplified and then sent to a speaker. The inner walls of the sound load chamber were made with sound-absorbing material. The outer walls of chamber were wrapped by a metal shell in order to reduce the influence of environmental noise. The waterproof speaker was immersed in the 75% alcohol for 30 min and then irradiated by UV to kill bacteria before it was put into the LB medium. The sound frequency and intensity was set according to the experiment design. More details were described in Gu et al. (2010).

Figure 1 Schematic of sound waves load apparatus.

(A) sound waves source; (B) sound waves transmission conductor; (C) speaker; (D) ultraviolet light; (E) beaker; (F) metal case; (G) sound-absorbing material; (H) magnetic stirrer.

Sound exposure experiments

Sound exposure test were performed in the experimental installations (Fig. 1). It was same with our previous equipment (Gu et al., 2010) except that the speakers were immersed in LB medium and the rotating sample holder was replaced by a magnetic stirrer. E.coli K-12 was exposed to different conditions: (I) sound frequency varied from 250 to 16,000 Hz and maintained sound intensity level 80 dB and sound power level 55 dB; (II) sound intensity level varied from 0 to100 dB and maintained sound frequency 8 kHz and sound power level 55 dB; (III) sound power level varied from 55 to 63 dB and maintained 8 kHz and 80 dB. Sound frequency and intensity level were adjusted by waveform generator and the amplifying circuit in the sound-wave generating unit, respectively. Sound intensity also known as acoustic intensity is defined as the sound power per unit area. Sound intensity level (SIL) is the level (a logarithmic quantity) of the intensity of a sound relative to a reference value. Sound power or acoustic power is the rate at which sound energy is emitted, reflected, transmitted or received, per unit time. Sound power level (SWL) is a logarithmic measure of the power of a sound relative to a reference value. The variation of sound power level was realized by adjusting the size of the speaker. Sound power level was measured by an instrument (LAN-XI, B & K, Denmark).

Samples without sound exposure served as a control group. The temperature within the sound waves load apparatus was maintained at 37 ± 1 °C. The sound exposure was performed continuously in the whole experiment, and the magnetic stirrer was stirring for 5 min per 15 min.

Measurement of biomass and specific growth rate

The biomass of E. coli K-12 was represented by maximum optical density. Optical density of culture broth was measured at 600 nm using a spectrophotometer (UV754N; Shanghai Aucy Scientific Instruments, Shanghai, China). Cell dry weight was performed according to the method of being dried for six hours at 70 °C , and the specific growth rate µ was calculated as follow: μ=Δmm×Δt

were m is the whole dry cell weight, μ is the specific growth rate, Δm is the addition of dry cell weight in Δt hours, and μmax is the maximum specific growth rate.

Measurement of E. coli K-12 intracellular protein and RNA

The culture was sampled every 6 h, and then concentrated or diluted to 1 (OD600). Protein was extracted using Bacterial Protein Extraction Kit (BS596, Sangon Biotech Co, Ltd, Shanghai, China) and quantified by a Modified BCA Protein Assay Kit (SK3051; Sangon Biotech Co, Ltd, Shanghai, China). Total RNA was extracted by HiPure Bacterial RNA Kit (R4181-01; Magen, China) and quantified by a spectrophotometer (DS-11; DeNovix, Wilmington, DE, USA).

Morphologic observation of E. coli K-12

Under sound wave frequency 8 kHz and power level 61 dB, E. coli K-12 was exposed to an intensity level of 80 dB and 100 dB, respectively. Cells were sampled at 48 h, centrifuged, washed with distilled water, dehydrated using graded ethanol (20%, 50%, 80%, 100%), and then dissoved in distilled water. Samples were dried on glass slides, and then a layer of metal film was plated on the surface of glass slide in a vacuum evaporator. Morphology observation of E. coli K-12 was performed by scanning electron microscope (SEM) (JSM-5610LV; JEOL, Tokyo, Japan). A total of 100 randomly selected bacterial cells were measured, and the average size of the bacterial cells was calculated.

Statistical analysis

All the experiments were performed in triplicate, and measurements are reported as mean ± standard deviation (SD). Statistical analysis was performed by applying variance (ANOVA) multiple comparisons (single factor) in SPSS. Treatment effects were considered to be significant at P < 0.05.

Results

Effect of different acoustic parameters on the growth of E. coli K-12

Effects of sound frequency on E. coli K-12

The effects of sound frequency on the biomass and μmax of E. coli K-12 were shown in Fig. 2. The results indicated that the sound treatment with different frequencies significantly increase the biomass of E. coli K-12. Significant differences (P < 0.001) in biomass were observed when E. coli K-12 was exposed to sound frequency 2 kHz and 8 kHz, which were increased by about 21.04% and 27.06% versus the control group, respectively. Meanwhile, exposure of E. coli K-12 to 2 kHz and 8 kHz sound waves also led to an increase of the μmax, reflecting a faster growth of the treated group than the control group. The μmax of the treated E. coli K-12 with 2 kHz and 8 kHz were 1.951 h−1 and 1.961 h−1 respectively. The behavior of the treated E. coli K-12 strongly suggested that sound waves exposure accelerated the E. coli K-12 growth and the biological effects induced by sound waves stimuli had a non-linear relationship with frequency, and showed obvious frequency peculiarities.

Figure 2 Effects of sound frequency on the growth of E. coli K-12.

All experiments were exposed to sound intensity level 80 dB and power level 55 dB. Asterisks indicate significance: ∗∗∗p < 0.001, ∗∗0.001 < p < 0.01, ∗0.01 < p < 0.05. Vertical bars represent means ± SD.

Effect of sound intensity level on E. coli K-12

Under frequency 8 kHz and power level 55 dB, E. coli K-12 was exposed to sound waves with different sound intensity levels. We found that the biomass of the E. coli K-12 were significantly higher in the treated group with sound intensity level 80 dB compared to the control group. A rapid increase of biomass in the treated group was observed reaching a maximum of 1.371 (OD600) with sound intensity level 80 dB, about 27.06% higher in the treated group as compared with the control group, and then it decreased sharply (Fig. 3). The μmax of E. coli K-12 increased sharply and reached the peak at sound intensity level 80 dB, and then enhanced more slowly (Fig. 3). Particularly when the sound intensity level was 100 dB, the μmax (2.151 h−1) was approximately 1.4 times that of the control group (1.562 h−1). Moreover, we also found that the logarithmic phase in the experimental group exposed to sound intensity level 80 dB was extended by 21.12% compared to the 100 dB sound stimuli (data was shown in Fig. S1). This is the reason why the μmax enhanced slowly but the biomass dramatically reduced, when the sound intensity was increased from 80 dB to 100 dB.

Figure 3 Effects of sound intensity level on the growth of E. coli K-12.

All experiments were exposed to the sound fields (sound intensity level varied from 0 to 100 dB and maintained sound frequency 8 KHz and sound power level 55 dB). Asterisks indicate significance: ∗∗∗p < 0.001, ∗∗0.001 < p < 0.01, ∗0.01 < p < 0.05. Vertical bars represent means ± SD.

Figure 4 Effects of sound power level on the growth of E. coli K-12.

All experiments were exposed to the sound fields (sound power level varied from 55 to 63 dB and maintained sound frequency 8 KHz and sound intensity level 80 dB). Asterisks indicate significance: ∗∗∗p < 0.001, ∗∗0.001 < p < 0.01. Vertical bars represent means ± SD.

Effect of sound power level on E. coli K-12

From Fig. 4, we observed that the growth of E. coli K-12 exposed to different sound power greatly increased. The biomass presented an approximate linear growth with the increase of sound power and reached the peak value at 59 dB, while it displayed slowly increasing from sound power level 59 dB to 61 dB and then reduced sharply. The maximum biomass of E. coli K-12 treated with sound power level 61 dB was 1.863 (OD600), about 1.7 times that of the control group (OD600 1.079). The μmax of E. coli K-12 was elevated rapidly to the peak at 61 dB, and then it declined drastically. Particularly when E. coli K-12 were exposed to sound power level 61 dB, the μmax (3.837 h−1) was about 2.5 times that of the control group (1.562 h−1). While the sound power level exceeded 61 dB, both biomass and μmax became gradually decreased, which could reflect that excess sound exposure might evoke an inhibition of the growth of E. coli K-12 by some potential mechanisms.

Effects of sound exposure on intracellular macromolecular synthesis in E. coli k-12

As shown in Fig. 5, we studied the effects of sound waves on the intracellular macromolecular of E. coli K-12 with frequency 8 kHz, intensity level 80 dB and power level 61 dB and found that certain sound exposure significantly affected the intracellular protein and RNA in E. coli k-12. The intracellular protein and RNA both in the treated group and the control group reduced slowly with time. Under sound exposure, the concentration of intracellular protein presented a significant increase in the treated group at 6 h, and the value of intracellular protein in the treated group reached 566.4 mg/g, about 1.1 times that of the control group (511.1 mg/g). The concentration of the intracellular RNA of E. coli K-12 also increased significantly in the treated group at 6 h. When E. coli K-12 were continuously exposed to sound waves for 6 h, the intracellular RNA (113.0 mg/g) in treated group was about 1.25 times that of the control group (90.1 mg/g). We concluded that sound exposure can significantly promote synthesis of the intracellular protein and RNA of E. coli K-12 in the early treatment stages, which was in favor of cell division.

Figure 5 The total intracellular protein and RNA of E. coli K-12 exposed to sound wave at different time.

(A) The total intracellular protein. (B) The total intracellular RNA. All experiments were exposed to sound frequency 8 KHz, intensity level 80 dB and power level 61 dB. Asterisks indicate significance: ∗∗∗p < 0.001, ∗∗0.001 < p < 0.01. Vertical bars represent means ± SD.

Morphological change of E. coli K-12 cell exposed to sound waves

E. coli K-12 cellular morphology was observed after sound exposed 48 h (Fig. 6). The length and width of E. coli K-12 cell was measured with the software carried by SEM. It was found that the average length of E. coli K-12 reached 2.060 ± 0.485 µm (80 dB) and 2.395 ± 0.904 µm (100 dB) respectively, and its length increases more than 27.26% under sound intensity level 100 dB compared to the control group (1.882 ± 0.375 µm). However, no difference was observed in width.

Figure 6 The bacterial cell morphology of E. coli K-12 of SCE.

(A) The cells in the control group. (B) The cells exposed to sound frequency 8 KHz, intensity level 80 dB and power level 61 dB. (C) The cells exposed to sound frequency 8 KHz, intensity level 100 dB and power level 61 dB. Cells were sampled at 48 h.

Discussion

Sound is a mechanical wave that results from the back-and-forth vibration of the particles of a medium. If it is moving through living organisms, then cells will be displaced both rightward and leftward as the energy of sound wave passes through them result in biological effects. Some organisms might respond to sound stimulation with a positive effect on growth. From Fig. 2, we noted that sound stimulation evidently promoted the growth of E. coli K-12. Cai et al. (2014) reported that the germination period of mung beans was reduced after audible sound treatments with 1.0–2.5 kHz. The PO algae under exposure of sound waves with frequency of 2,200 Hz had greatly significant increase in dry biomass (Cai et al., 2013). Chen (2013) also showed that sound waves with main frequency such as 2 kHz in environment of wild plants had better effects on plant growth than other kinds of audible sound. However, the inhibition effect of sound waves on microbial growth was also observed. Sarvaiya & Kothari (2015) reported that Serratia marcescens were found to suffer a decrease in growth under the influence of music. Also, all sonic stimuli tested reduced biomass production of the yeast cells by 14% (Aggio, Obolonkin & Villas-Bôas, 2012). In addition, we also found that the sound waves at frequency 2 kHz and 8 kHz evoke the most significant growth promotion of E. coli K-12. It suggested that the action of sound waves showed obvious frequency peculiarities. Matsuhashi et al. (1998) found that cells of B. Subtilis could produce sound waves and the frequencies of the sound produced by B. Subtilis were similar with the frequencies that induced a response in B. Carboniphilus. Furthermore, microbial cells can absorb more energy when the frequency of the incoming vibration matches their natural frequency of vibration (Reguera, 2011). Ying, Dayou & Chong (2009) revealed that the increases of viable cells of E. coli were equivalent to 7%, 34% and 30.5% for the sound treatment at 1 kHz, 5 kHz and 15 kHz waves, compared to the control group. Our previous work found that the biomass (OD600) of E. coli N43 increased more than 31.5%, 86.0% and 31.1% for the sound treatment at 1 kHz, 5 kHz and 10 kHz than the control group. In this paper, we found that the biomass of E. coli K-12 increased more than 13.3%, 15.3% and 6.8% for the sound treatment at 1 kHz, 4 kHz and 10 kHz than the control. These findings suggested that significant differences of biological effects induced by sound stimulation exist between different strains of the same species, species of the same genus and genera of the same family.

The investigations of sound waves exposed to different intensity and power level showed that sound stimulation at certain strength can promote the growth of E. coli K-12. Sun & Cai (1999) and Shen et al. (1999) observed that sound stimulation could benefit the absorb of nutriment and synthesis of DNA in S period of tobacco cells and promote the fluidity of membrane wall and membrane lipid. However, our experiments indicated that a high level of sound power (sound power level at 63 dB or sound intensity at 100 dB) could induce an obvious decrease in E. coli K-12 growth promotion effects. Li et al. (2001) noted that when the sound intensity increase from 100 dB to 110 dB, the number of tobacco cells in S period reduced greatly. Consequently, excessive sound exposure might bring out negative effects on the growth of E. coli K-12 through some unknown way.

It is necessary for cell division to accumulate rapidly intracellular biological macromolecules, such as nucleic acid molecules, proteins, lipids and polysaccharides. As shown in Fig. 5, sound exposure could significantly promote synthesis of the intracellular protein and RNA in the early treatment stages. The value of intracellular protein and RNA at 6 h reached 566.4 mg/g and 113.0 mg/g in the treated group were 1.1 times and 1.25 times that of the control group respectively. It was also reported that sound stimulation can promote the synthesis of intracellular molecules such as protein (Yang, 2013), RNA (Wang et al., 2003) and DNA (Li et al., 2001) in plants. Schaechter, Maaloe & Kjeldgaard (1958) authenticated that not only cell mass, but also nucleic acid and protein content were a function of growth rate rather than the composition of the medium used to achieve that growth rate.

In addition, Vadia & Levin (2015) verified that cell size is a linear function of growth rate. Taheri-Araghi et al. (2014) demonstrated that cells add a constant volume each generation based on the combination of experimental results and quantitative analysis. In our experiments, sound stimulation can induce the change of cell morphology except for the growth, metabolism and cell division. We found that the average length of E. coli K-12 exposed to sound intensity level 100 dB increased more than 27.26% compared to the control group. Pelling et al. (2004) observed that the cell wall of cells of Saccharomyces cerevisiae exhibited local, periodic nanoscale motions in an acoustically insulated environment using an atomic force microscope (AFM). The membrane fluidity increased under sound stimulation of some strength and frequency (Zhao et al., 2002). Sound stimulation also changed the secondary structure of the cell wall proteins of tobacco cells.

The exact mechanism of bacterial cells respond to sound exposure is unclear. Mechanosensitive channels (Msc) might be involved in the process of acoustic stress. Actually, Msc play an important role in mechanical signal transduction. It is widely distributed in cell membrane of bacteria, which was considered as cell “sensor” and “effector” for mechanical stimulation (Anishkin & Kung, 2005; Martinac, 2011). When sound waves got through the cell membrane surface it would open Msc, resulting in some small molecules outside the cell such as H2O, Na+, K+ and Ca2+ pass freely through the cell membrane (Booth et al., 2007; Nazarenco et al., 2003). It is well known that Ca2+, as a second messenger, plays an important role in the life activities of microbes (Ren et al., 2009). Furthermore, studies of the plant response to mechanical stimulation found that calcium signal was an early event in stress reaction (Zhang et al., 2010). We speculated that bacterial cells might sense the sound stimuli by Msc and convert physical stimuli into biological signals by the inflow of Ca2+. In addition, bacteria dominated the increasing cell-population density by quorum sensing (Taj et al., 2014), and E. coli had a whole quorum sensing system with AI-2 as signal molecule (Ting, 2009). It was obvious that the cell-population density of E. coli K-12 increased greatly under the condition of sound stimuli and without the addition of any nutriments compared to the control group. The result suggested that sound exposure may assist E. coli K-12 in obtaining a relatively high threshold level of cell growth by passivated density—dependent inhibition. In the subsequent research, we will aim to explore and verify the related mechanism.

The results of this study indicated that sound while travelling through microbial suspensions created a kind of mechanical stress, which can be sensed by a growth vessel inside cells, and living organisms including microbes can rapidly respond to the stress at both the transcriptional and posttranscriptional levels. However, the mechanism of sound stimulation on microorganism growth is still unknown. Our further work will concentrate on the production of AI-2 and the concentration of intracellular calcium in E. coli K-12 that occurs in response to sound waves, which will certainly provide new insights into the interaction of microbes with sound exposure in general.

Supplemental Information

Data S1 Raw data

Click here for additional data file.

Figure S1 Effect of sound intensity on the logarithmic phase of E. coli K-12

All experiments were stimulated at sound frequency 8 KHz and power 55 dB. Asterisks indicate significance: ∗∗0.001 < p < 0.01. Vertical bars represent means ±SD.

Click here for additional data file.

Additional Information and Declarations

Competing Interests

Author Contributions

Data Availability

The authors declare there are no competing interests.

Shaobin Gu conceived and designed the experiments, analyzed the data, reviewed drafts of the paper.

Yongzhu Zhang performed the experiments, analyzed the data, contributed reagents/materials/analysis tools, wrote the paper, prepared figures and/or tables.

Ying Wu conceived and designed the experiments, performed the experiments, analyzed the data, contributed reagents/materials/analysis tools, wrote the paper, reviewed drafts of the paper.

The following information was supplied regarding data availability:

The raw data has been supplied as Data S1.

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
