# Peer review of "Effects of sound exposure on the growth and intracellular macromolecular synthesis of E. coli k-12"

_PeerJ, doi:10.7717/peerj.1920_

## Round 0.1 · original submission · Major Revisions

· Academic Editor

Major Revisions

Please take account of the comments of both reviewers. Please make sure to look carefully at the annotated manuscript kindly provided by the reviewer in your resubmission.

I also urge you to provide strong reasons as to why this study was performed in the first place and some mechanistic insight would aid us in making a final decision on the manuscript

·

Basic reporting

Effect of sound exposure on the growth and intracellular macromolecular synthesis of E. coli k-12 by Gu et al., reports on the effect of sound waves on the growth of E.coli K12.

This reports talks about the use of sound waves of different Hz and Db on E.coli and looking into its growth and intracellular macromolecular synthesis.

At present the study is phenomenon based and needs answering crucial mechanisms.

Experimental design

An elaborate review by Gemma Reguera in Trends in Microbiology, 2011 ( 19,105-113) talks about excellent papers on the effect of sound in various model system. Hence, this paper becomes redundant as this is not a new report.

The experimental design should take into consideration different media composition with different viscosity as sound travel different in different medium

Validity of the findings

Though it is very nice to see the effect of any system on organism, at present I am not able to appreciate this study as it does not brings out why this study was done.

Additional comments

This study talks about the biomass and the macromolecular synthesis in E.coli in response to sound waves.

1. How is the signal perceived by E.coli?

2. The EM data adds nothing to the study. AFM is required to understand the intricate changes int he membrane or cytosol.

3. The increase in biomass will lead to increased synthesis of macromolecules. This experiment also does not add up anything to the study.

4. Is this phenomenon conserved among different strains of E.coli?

·

Basic reporting

Authors have chosen to investigate in the area of cell-sound interaction. This is an interesting, but much less-researched area. Hence any meaningful contribution in this area of research should be encouraged.

Experimental design

Experimental design seems to be sound. However, the manuscript needs to include some explanatory notes on, how sound 'intensity' differs from 'power'. By providing these (and other such) details, the authors can make their manuscript more readable for a audience, which is likely to be comprised largely of biologists.

Validity of the findings

Findings seems to be acceptable.

Additional comments

Please see the comments inserted in the manuscript file itself.

---

## Round 0.2 · Minor Revisions

· Academic Editor

Minor Revisions

There appear to be some issues that have not been addressed based on the comments of Reviewer 2. While minor, they must be addressed before acceptance.

·

Basic reporting

No comments

Experimental design

No comments

Validity of the findings

No comments

·

Basic reporting

No new comments.

Experimental design

no new comments

Validity of the findings

My original comment on figure-4, regarding high OD values seems to have remained unanswered. However, I leave it now to the Editor to make final decision on it.

Additional comments

"speakers were immersed in nutrient solution" in the revised version; this sentence is almost the same as "speakers were put into nutrient solution" of the original submission.
I still feel, it conveys wrong meaning. Was the speaker in direct contact of the liquid?
Please modify that sentence so as to leave no confusion for the reader.

---

## Round 0.3 · accepted · Accept

· Academic Editor

Accept

Minor revisions have been suitably made.